# Ultracold field-linked tetratomic molecules

Xing-Yan Chen[1,2], Shrestha Biswas[1,2], Sebastian Eppelt[1,2], Andreas Schindewolf[1,2], Fulin Deng[3,4], Tao Shi[4,5 ✉], Su Yi[4,5,6], Timon A. Hilker[1,2], Immanuel Bloch[1,2,7] & Xin-Yu Luo[1,2 ✉]

Ultracold polyatomic molecules offer opportunities[1] in cold chemistry[2,3], precision measurements[4] and quantum information processing[5,6], because of their rich internal structure. However, their increased complexity compared with diatomic molecules presents a challenge in using conventional cooling techniques. Here we demonstrate an approach to create weakly bound ultracold polyatomic molecules by electroassociation[7] (F.D. et al., manuscript in preparation) in a degenerate Fermi gas of microwave-dressed polar molecules through a field-linked resonance[8–11]. Starting from ground-state NaK molecules, we create around $1.1 \times 10^3$ weakly bound tetratomic $(NaK)_2$ molecules, with a phase space density of 0.040(3) at a temperature of 134(3) nK, more than 3,000 times colder than previously realized tetratomic molecules[12]. We observe a maximum tetramer lifetime of 8(2) ms in free space without a notable change in the presence of an optical dipole trap, indicating that these tetramers are collisionally stable. Moreover, we directly image the dissociated tetramers through microwave-field modulation to probe the anisotropy of their wavefunction in momentum space. Our result demonstrates a universal tool for assembling weakly bound ultracold polyatomic molecules from smaller polar molecules, which is a crucial step towards Bose–Einstein condensation of polyatomic molecules and towards a new crossover from a dipolar Bardeen–Cooper–Schrieffer superfluid[13–15] to a Bose–Einstein condensation of tetramers. Moreover, the long-lived field-linked state provides an ideal starting point for deterministic optical transfer to deeply bound tetramer states[16–18].

Molecules exhibit a rich set of internal and external degrees of freedom, which can be fully controlled only under ultracold temperatures (<1 mK) (refs. 19,20). For example, ultracold molecules prepared in well-defined quantum states enable studying quantum dynamics[21], chemical reactions with state-to-state control[20] and quantum scattering[3,11,22] at an unprecedented level. The highly tunable long-range interactions in dipolar molecules also give rise to many-body phenomena[23] such as exotic dipolar supersolids[24] and $p$-wave superfluids[13–15]. Furthermore, ultracold polyatomic molecules have emerged as a powerful platform for various applications, including tests of beyond-Standard-Model physics[4], non-equilibrium dynamics[25] and quantum information processing[5,6,26], because of their additional degrees of freedom compared with diatomic molecules.

Notable progress has recently been made in the field of molecular cooling, enabling quantum degeneracy in ultracold gases of diatomic dipolar molecules[27–29]. However, for larger molecules, reaching the ultracold regime remains challenging because of their increased complexity and adverse collisional properties. Direct cooling techniques such as buffer gas cooling[30], supersonic expansion[31], beam deceleration[32], cryofuges[33] and optoelectrical Sisyphus cooling[12] have only marginally reached ultracold temperatures. Laser cooling of larger polyatomic molecules is an area of active research[34,35]. Although symmetric top molecules have been laser-cooled in one dimension[36], it

remains to be seen how efficient laser cooling of large (tetratomic or larger) molecules will be in three dimensions and whether temperatures below the submicrokelvin regime can be achieved. Recently, magnetoassociation of ultracold molecules by Feshbach resonances has been extended to weakly bound triatomic $NaK_2$ molecules in the 100 nK regime[37], in which the molecules inherit the low temperature from the atom–diatomic molecule mixture. However, this technique requires resolvable Feshbach resonances between the collisional partners. For larger, polyatomic molecules, the high number of the intermediate collisional states and their fast loss mechanisms at short range results in a nearly universal collisional loss rate[38], preventing the occurrence of these Feshbach resonances.

Here we demonstrate a previously unknown and general approach to form weakly bound ultracold polyatomic molecules by electroassociation of smaller polar molecules[7] (F.D. et al., manuscript in preparation). We create ultracold tetratomic $(NaK)_2$ molecules from pairs of fermionic NaK molecules in microwave-dressed states by ramping the microwave field across a field-linked scattering resonance[8–11]. This approach benefits from the universality of field-linked resonances and can be applied to any molecule with a sufficiently large dipole moment. We measure a lifetime of up to 8(2) ms of our field-linked tetramers near the dissociation threshold and achieve a phase space density of 0.040(3). With microwave-field modulation dissociation after time of

[1]Max-Planck-Institut für Quantenoptik, Garching, Germany. [2]Munich Center for Quantum Science and Technology, Munich, Germany. [3]School of Physics and Technology, Wuhan University, Wuhan, China. [4]CAS Key Laboratory of Theoretical Physics, Institute of Theoretical Physics, Chinese Academy of Sciences, Beijing, China. [5]AS Center for Excellence in Topological Quantum Computation & School of Physical Sciences, University of Chinese Academy of Sciences, Beijing, China. [6]Peng Huanwu Collaborative Center for Research and Education, Beihang University, Beijing, China. [7]Fakultät für Physik, Ludwig-Maximilians-Universität, Munich, Germany. ✉e-mail: tshi@itp.ac.cn; xinyu.luo@mpq.mpg.de

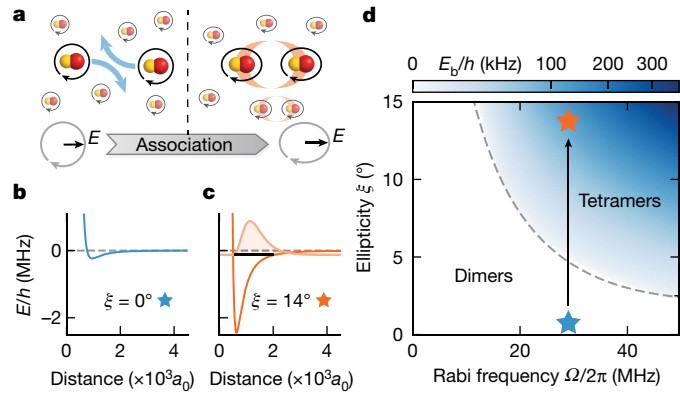

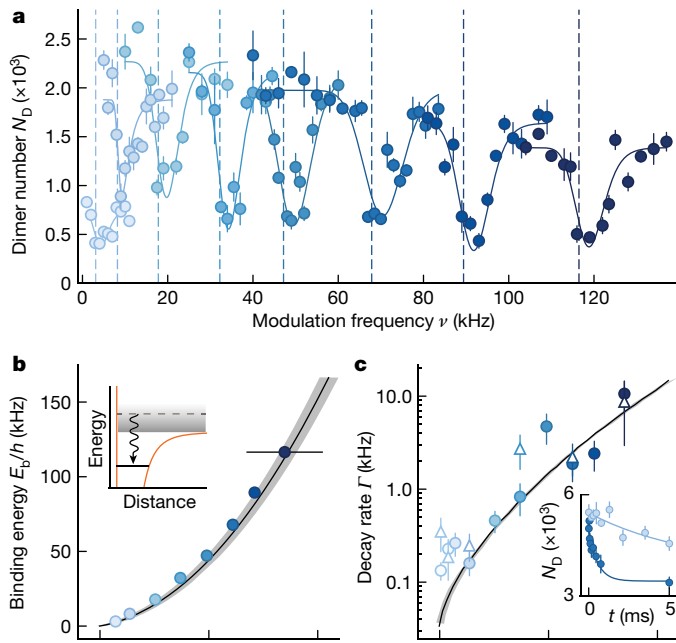

**Fig. 1 | Electroassociation of field-linked tetramers. a**, Microwave-dressed NaK dimers are associated into $(NaK)_2$ tetramers as the microwave polarization is ramped from circular to elliptical. **b,c**, Interaction potentials between two dimers approaching along the long axis of the microwave field. The potential depth increases with the ellipticity $\xi$ and a tetramer bound state emerges from the collisional threshold. The light orange line shows the radial wavefunction of the tetramer, and the black solid line indicates its binding energy. **d**, Calculated binding energy of the tetramers. The field-linked resonance (dashed line) marks the onset of the tetramer state. The stars and the arrow mark the electroassociation trajectory in the experiment. Within the range of experimental parameters, there exists only a single field-linked tetramer state (Methods).

flight, we directly image the tetramers and show the expected anisotropic angular distribution.

## Field-linked tetramers

A microwave field-linked molecule consists of two microwave-dressed polar molecules bound by long-range dipole–dipole interactions. Each constituent molecule is dressed by a near circularly polarized microwave field, which mixes different rotational states and induces a rotating dipole moment of up to $d_0/\sqrt{6}$ in the laboratory frame, where $d_0 \approx 2.7$ debye is the dipole moment of NaK in its body-fixed frame. The strong induced dipole–dipole interaction potential can host stable tetratomic bound states that give rise to scattering resonances[11]. By ramping the microwave field across these resonances, a pair of scattering NaK dimers can be adiabatically associated into a $(NaK)_2$ tetramer (Fig. 1a). We refer to this process as electroassociation[7], analogous to magnetoassociation using a magnetic Feshbach resonances[39].

The concept behind electroassociation involves a smooth transition from low-lying scattering states of a dimer pair to the bound tetramer state by gradually ramping the microwave field[7] (F.D. et al., manuscript in preparation). The increase in the microwave field ellipticity (Fig. 1b,c) enhances the depth of the interaction potential, leading to the emergence of the tetramer state from the collisional threshold and an increase in its binding energy (Fig. 1d). Moreover, microwave shielding of the dimers leads to enhanced collisional stability of the field-linked tetramers[40,41] (F.D. et al., manuscript in preparation), which can therefore be efficiently associated from a low entropy gas of dimers.

## Binding energy and lifetime

Our experiments begin with an ultracold gas of optically trapped (1,064 nm) ground-state $^{23}Na^{40}K$ molecules with nuclear spin projections $(m_{i,Na}, m_{i,K}) = (3/2, -4)$, which are formed from an ultracold atomic mixture by magnetoassociation and stimulated Raman adiabatic passage[28]. We subsequently dress the molecules with a circularly polarized microwave field, blue detuned to the transition between the ground and the first rotational excited states, to shield the molecules

**Fig. 2 | Tetramer binding energy and lifetime. a**, Tetramer association spectra at different ellipticities obtained by modulating the ellipticity of the microwave field. The solid lines show the fitted line shape, and the dashed lines mark the extracted binding energies. The line shapes are shifted and broadened by the linewidth of the tetramer states and other technical broadening effects (Methods). The Rabi frequency of the microwave field is $\Omega = 2\pi \times 29(1)$ MHz and detuning $\Delta = 2\pi \times 9.5$ MHz. The peak-to-peak modulation amplitude is 1° and the modulation time is 100 ms, except for the lowest ellipticity for which we use an amplitude of 0.5° and a modulation time of 400 ms. The error bars represent the standard error of the mean of four repetitions. **b**, Binding energy $E_b$ obtained from the association spectra (circles) compared with theory prediction (line). The statistical error bars are smaller than the symbol size. The black error bar marks the systematic uncertainty of ellipticity. The shaded area shows theoretical calculations, including the systematic uncertainty of the Rabi frequency $\Omega$. The inset illustrates the radiofrequency association from free to bound states. **c**, Decay rate $\Gamma$ of the tetramers in time of flight (circle) and in trap (triangle), compared with theory calculations (line). The error bars show the fitting errors. The inset shows example decay curves at $\xi = 7(1)°$ and $\xi = 11(1)°$ in time of flight. The error bars represent the standard error of the mean of eight datasets.

from two-body collisions and perform evaporative cooling[42]. Depending on the trap depth at the end of the evaporation, we prepare various initial conditions of the molecular gas. The minimum temperature is $T = 50(1)$ nK at a dimer molecule number $N_0$ of $5.7(3) \times 10^3$, corresponding to $T/T_F = 0.44(1)$, where $T_F$ is the Fermi temperature of the trapped gas. The trapping frequencies are $(\omega_{\tilde{x}}, \omega_{\tilde{y}}, \omega_z) = 2\pi \times (42, 61, 138)$ Hz, where $z$ is the vertical direction.

We probe the binding energy of the tetramers using microwave-field modulation association spectroscopy. We start the experiment with a circularly polarized microwave field at a Rabi frequency $\Omega = 2\pi \times 29(1)$ MHz and detuning $\Delta = 2\pi \times 9.5$ MHz (ref. 42). We then quickly ramp the microwave in 100 μs to a target ellipticity $\xi$ above the field-linked resonance and modulate the ellipticity at various frequencies for up to 400 ms. The ellipticity $\xi$ is defined such that $\tan\xi$ gives the ratio of the left- and right-handed circularly polarized field components. When the modulation frequency $\nu$ is slightly above the binding energy, tetramers are formed and subsequently decay into lower dressed states accompanied by a large release energy. This leads to a reduction in the remaining dimer number, which we detect in the experiment. As shown in Fig. 2a, we observe clear asymmetric line

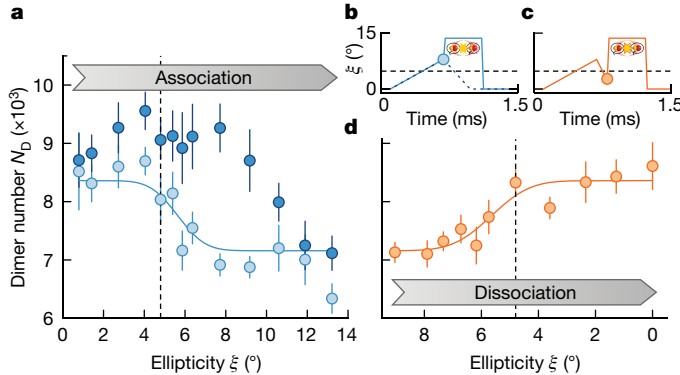

**Fig. 3 | Association and dissociation processes. a**, Remaining dimer number $N_D$ after the association ramp. The dark blue circles show the total number of dimers, including dissociated tetramers, whereas the light circles show the dimer number after removal of the tetramers. The solid blue line is a fit to an error function. The vertical dashed line marks the theoretical resonance position. **b,c**, Waveform of the association (**b**) and dissociation (**c**) ramps. In **b**, the blue solid line shows the waveform with the removal of the tetramers and the blue dashed line shows the waveform without the removal of the tetramers. The horizontal dashed lines indicate the theoretically predicted resonance position. The circles show the target ellipticity of the association or dissociation ramp, which is plotted in **a** and **d**, respectively. **d**, Increase in the detected dimer number during the dissociation ramp. The solid orange line is a fit to an error function. The vertical dashed line shows the predicted resonance position. The error bars represent the standard error of the mean of ten experiment repetitions.

shapes in the spectra, in which the onset frequency of the tetramer association corresponds to the binding energy of the tetramer (Methods).

Next, we probe the lifetime of the tetramers by measuring their loss dynamics. The dominant loss process for tetramers is spontaneous dissociation into lower microwave-dressed states[40] (F.D. et al., manuscript in preparation) accompanied by a large gain in kinetic energy, which, effectively, leads to a one-body decay of the tetramer number. To investigate this process, we first create tetramers by ramping the ellipticity to $\xi = 8(1)°$ in 0.67 ms, and then quickly ramp to a target ellipticity in 20 μs. There we hold for a variable time, then reverse the ellipticity ramps to dissociate the tetramers back to dimer pairs to map the loss of tetramers during the hold time onto the total dimer number. We turn off the trap after the association to minimize collisional loss. We observe a fast initial decay, in which the decay rate increases with the binding energy of the tetramers (Fig. 2c, inset). These initial decays are much faster than the expected dimer–dimer collisional loss rates and are absent if we jump from $\xi = 0°$ to the target ellipticity, so that no tetramers are expected to form. We, therefore, attribute this initial decay to the one-body loss of the tetramers. A maximum of 8(2) ms lifetime is observed near the dissociation threshold. With higher Rabi frequencies and at circular polarization, theory predicts lifetimes in excess of 100 ms at $E_b < h × 4$ kHz, where $h$ denotes the Planck constant (Methods). Additional measurement of lifetimes in the trap suggests that tetramers are collisionally stable against collisions with dimers or other tetramers (Methods).

## Association and dissociation processes

We probe the association and dissociation processes by ramping the ellipticity starting from $\xi = 0°$ with a constant ramp speed of 14° ms$^{-1}$ (27° ms$^{-1}$ for the dissociation) to a target ellipticity (Fig. 3b,c). To distinguish the tetramers from the unpaired dimers, we selectively remove the tetramers from the dimer–tetramer mixture by quickly ramping the ellipticity to $\xi = 14(1)°$ in 20 μs and hold for 0.4 ms, in which the tetramers are deeply bound and rapidly decay. Figure 3a shows that

the number of unpaired dimers (light blue) reduces as we ramp the ellipticity across the field-linked resonance, indicating tetramer formation, whereas Fig. 3d shows that the number of detected dimers revives as we ramp back to circular polarization, indicating that the formed tetramers can be reversibly dissociated back into dimer pairs. Moreover, we characterize the association process without removing the tetramers but followed by a dissociation ramp back to $\xi = 0°$. The detected dimer number (dark blue in Fig. 3a) partially revives until $\xi \gtrsim 12°$, in which the tetramers decay during the ramps before they can be dissociated back into dimers.

We experimentally optimize the efficiency of electroassociation by varying the ramp speed and the initial degeneracy of the dimers, achieving a maximum 25% conversion efficiency (Methods).

## Imaging of the dissociated tetramers

We use two methods to obtain absorption images of the tetramers. First, we image the adiabatically dissociated tetramers in time of flight to directly probe their temperature. The images of the tetramer momentum distribution are obtained by subtracting images without from images with the removal of tetramers at high ellipticity. Examples of such tetramer images are shown in Fig. 4a. From a fit to these time-of-flight images and considering the mass of the particles, we determine the temperature of the tetramers to be 134(3) nK, which is slightly higher compared with the dimer temperature of 97(6) nK. The fact that the tetramer cloud is smaller than the dimer background suggests partial thermalization and therefore elastic scattering during the electroassociation. Beyond that, heating might occur during the association and dissociation processes. From the number and trapping frequencies, we obtain a peak density of $5.0(2) × 10^{11}$ cm$^{-3}$ and a phase space density of 0.040(3) in the trap. We consider only the statistical error in this analysis.

Second, we image modulation-dissociated tetramers to probe the angular distribution of their single-particle wavefunction[43]. Here we modulate the ellipticity at a modulation frequency $v > E_b/h$, which couples the tetramer states to the scattering continuum. The coupled scattering state possesses a large wavefunction overlap with the tetramer state and preserves its angular distribution (Methods).

We take the difference between images with and without modulation to obtain images of the dissociated tetramers. Figure 4b shows the dissociation spectrum, which demonstrates an increase in the observed dimer number $N_D$ caused by the presence of dissociated tetramers when the modulation frequency $v$ exceeds the frequency associated with the binding energy of the tetramer $E_b/h = 17.8(3)$ kHz. At higher frequencies, $N_D$ declines because of a decrease in dissociation efficiency resulting from the diminished Frank–Condon factor, and the size of the dissociation pattern increases because of the higher dissociation energy. As shown in Fig. 4c,d, the dissociation pattern has two lobes, which are oriented along the long axis $x$ of the microwave polarization and match qualitatively with the theoretical wavefunction in Fig. 4e,f. Radial integration of the image shows the angular distribution of the wavefunction, which follows $p$-wave symmetry[44] in the $p_x$ channel $\cos^2\phi$, where $\phi$ is the angle from the $x$-axis (Methods). The broken rotational symmetry along the quantization axis is a result of the elliptical microwave polarization. When we rotate the microwave field by roughly 90°, by flipping the sign of the relative phase between the two feeds of the antenna (Methods), the dissociation pattern is similar but rotated by about 90°, which demonstrates the tunable control of the tetramer wavefunction through the microwave field.

## Discussion

By efficient electroassociation in a degenerate Fermi gas of diatomic molecules, we have created a gas of field-linked tetramers at unprecedentedly cold temperature. The associated weakly bound (NaK)$_2$

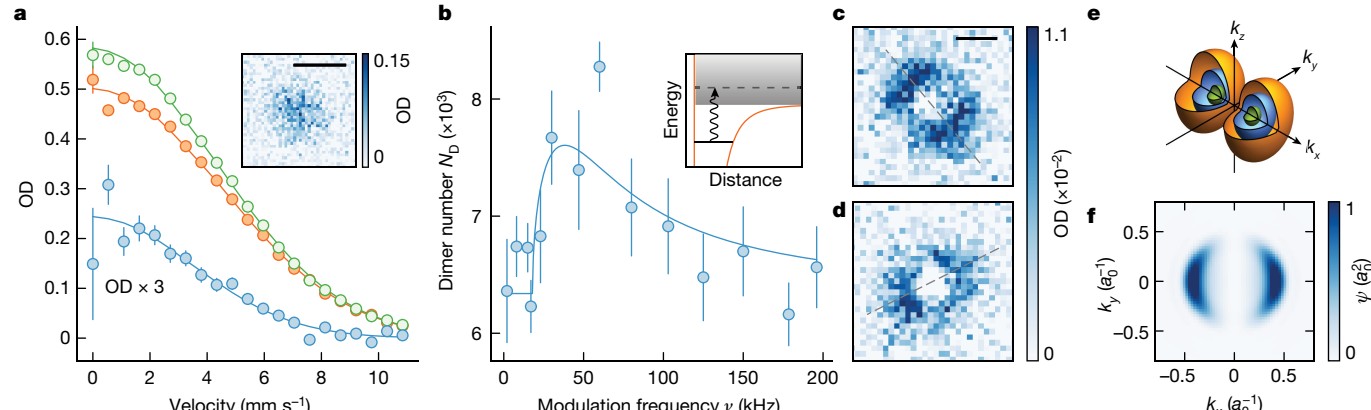

**Fig. 4 | Momentum distributions of the dissociated tetramers. a,** Azimuthally averaged optical density (OD) of the samples after ramp dissociation and 4.5 ms time of flight. The difference (blue) between images with (orange) and without (green) the removal of tetramers shows the momentum distribution of the tetramer cloud. The error bars represent the standard error of the mean of 60 repetitions. The inset shows the difference image. **b,** Tetramer dissociation spectrum. We create the tetramers at $\xi = 8(1)°$ and modulate the ellipticity with an amplitude of 1.4° for 2 ms. The solid line is a fit to the dissociation line shape (Methods). The error bars represent the standard error of the mean of ten repetitions. **c,d,** Time-of-flight images of modulation-dissociated tetramers. We use a modulation frequency of 30 kHz, with an amplitude of 3.6° for 2 ms. Although the microwave ellipticity is about the same in **c** and **d**, the field

orientation differs by about 90°. The dashed lines mark the extracted long axes of the patterns (Methods). The images are averaged over 84 and 40 measurements for **c** and **d**, respectively. Each pixel is a binning of 5 × 5 pixels from the raw images. **e,** Theoretical tetramer wavefunction in momentum space. The microwave field propagates along the $z$-axis, and its long axis is oriented along the $x$-axis. The cut-open surfaces correspond to a probability density of $1.5 \times 10^8 a_0^3$ (orange), $3.5 \times 10^8 a_0^3$ (blue) and $6 \times 10^8 a_0^3$ (green), respectively. **f,** The theoretical wavefunction of the dissociated tetramers in momentum space, integrated along the propagation axis of the microwave field. The imaging plane (**a,c,d**) is roughly perpendicular to the $z$-axis. Scale bars, 50 μm (**a**) and 100 μm (**c,d**).

molecules are more than 3,000 times colder than any other tetratomic atomic molecules produced so far, to our knowledge[12]. The created tetramers possess a phase space density of 11 orders of magnitude higher than the previous record, which is only two orders of magnitude below the quantum degeneracy. Starting below the critical temperature of $0.14T_F$, we expect a tetramer Bose–Einstein condensation (BEC) to emerge from a degenerate Fermi gas of dimers[45], realizing a Bardeen–Cooper–Schrieffer (BCS)–BEC crossover[46] that features anisotropic pairing because of the dipolar interactions[15].

The creation of field-linked tetramers creates an opportunity for exploring the rich landscape of the four-body potential energy surfaces (PESs). Similar to diatomic molecules, the long-lived weakly bound field-linked state provides an ideal starting point for deterministic optical transfer to deeply bound states within the PES[16,18]. For the PES of $(NaK)_2$ molecules, there are three energy minima that feature distinct geometries, including $D_{2h}$, $C_s$ and $C_{2v}$ symmetries[17]. These states possess electric dipole and/or quadruple moments, and together with their rich rovibrational structures, open up possibilities for studying eight-body collisions and quantum many-body phenomena with both strong dipolar and quadrupolar interactions.

The demonstrated electroassociation using field-linked resonances is applicable to any polar molecule with a sufficiently large dipole moment[7,10,41,47,48]. For example, it can be applied to laser-cooled polyatomic molecules, such as CaOH and SrOH (ref. 10), to form hexatomic molecules and beyond (Methods). Electroassociation can be generalized to d.c. electric fields, in which interspecies field-linked resonances could enable the association of two molecules from distinct molecular species. We can even imagine a scalable assembling process, in which we sequentially associate pairs of deeply bound molecules into weakly bound field-linked molecules, convert them into deeply bound states by optical transfer[16,18] and associate these molecules into even larger field-linked molecules.

## Conclusion

We have created and characterized field-linked tetratomic $(NaK)_2$ molecules, which are so far the first tetratomic molecules attained

in the 100 nK regime, to our knowledge. The properties of these tetramers are highly tunable with the microwave field and can be sufficiently long-lived and collisionally stable. Owing to the universality of field-linked resonance, our approach can be generalized to a wide range of polar molecules, including more complex polyatomic molecules. Our results provide a general approach to assemble weakly bound ultracold polyatomic molecules and open up possibilities to investigate several quantum many-body phenomena.

During the completion of this work, we became aware of a related theoretical proposal on the electroassociation of field-linked tetramers from bosonic dimers[7].

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

## Methods

### Microwave setup

The microwave setup is described in detail in ref. 11. We use a dual-feed waveguide antenna capable of synthesizing arbitrary polarization using two independent controllable feeds. The finite ellipticity and interference between these feeds result in an observed change in Rabi frequency of approximately ±4% when adjusting the relative phase, thereby contributing to the systematic uncertainty.

In terms of the control electronics, we have upgraded the amplifiers to 100 W (Qualwave QPA-5600-5800-18-47, the gate voltage of which is provided by a custom-made linear power supply) to achieve higher Rabi frequencies. Moreover, we have implemented filter cavities to suppress phase noise. Furthermore, we have incorporated a voltage-controlled phase shifter, enabling dynamic control of the relative phase between the two feeds for fine-tuning the microwave ellipticity. To maintain a constant output power while adjusting the ellipticity, we monitor the power in each feed using a power detector and use a feedback control using a voltage-controlled attenuator.

### Dimer loss near the field-linked resonance

We experimentally map out the field-linked resonance by measuring the dimer loss. Extended Data Fig. 1 shows the remained dimer number after a 100-ms hold time at $\Omega = 2\pi \times 29(1)$ MHz, $\Delta = 2\pi \times 9.5$ MHz, as a function of ellipticity $\xi$. The loss dip position matches the theoretical resonance position $\xi = 4.8°$.

### Conditions for efficient electroassociation

We experimentally identify the optimum condition for electroassociation. We obtain the tetramer number from the difference between images with and without the tetramer removal process outlined previously. First, we probe the timescale of the tetramer formation. We ramp the ellipticity from $\xi = 0(1)°$ to $8(1)°$ and vary the ramp speed. As shown in Extended Data Fig. 2a, we observe the formation of tetramers within 0.3(1) ms and subsequently decay because of the finite lifetime. We estimate that the tetramers scatter on average more than once during the association, bringing them close to thermal equilibrium with the remaining dimers.

Next, we investigate the role of quantum degeneracy in efficient electroassociation. For magnetoassociation of Feshbach molecules, it has been shown that a low entropy sample is crucial to achieve high conversion efficiency, because of the improved phase–space overlap between the atoms[49]. Here we vary the degeneracy of our initial dimer samples by changing the final trap depth of the evaporation[42]. We observe an increase in the conversion efficiency $\eta$, that is, the fraction of dimers converted into tetramers, with quantum degeneracy of the dimer gas. We achieve a maximum $\eta = 25(2)\%$ conversion efficiency at $T = 0.44(1)T_F$. Similar to that for magnetoassociation[49], a maximum unity conversion efficiency is expected at zero temperature.

### Association spectra analysis

We determine the binding energy of the tetramers for different target ellipticities (Fig. 2b) and find excellent agreement between the experimental data and coupled-channel calculations without free parameters.

We assume the dimer loss in the modulation spectra to be proportional to the number of formed tetramers. The line shape can be modelled using Fermi's golden rule[50]

$$N_T(\nu) \propto \int_0^\infty d\epsilon_r F(\epsilon_r) g(\epsilon_r) e^{-(h\nu - E_b - \epsilon_r)^2/\sigma^2} \qquad (1)$$

where $\nu$ is the modulation frequency and $E_b$ is the binding energy of the tetramer. The function $g(\epsilon_r) \propto e^{-\epsilon_r/k_B T}$ denotes the number of colliding pairs per relative kinetic energy interval $d\epsilon_r$. Here, the

temperatures $T$ are obtained from the data located away from the association transitions. The function $F(\epsilon_r) \propto \sqrt{\epsilon_r}(1 + \epsilon_r/E_b)^{-2}$ denotes the Franck–Condon factor $F(\epsilon_r)$ between the unbound dimer state and the tetramer state, which we assume to take the same form as for Feshbach molecules[50]. The product $F(\epsilon_r)g(\epsilon_r)$ is convoluted with a Gaussian distribution with the width $\sigma$ to account for the linewidth of the tetramer state and the finite energy resolution. The extracted linewidth shows a similar trend with ellipticity as the theoretical linewidth but slightly larger.

### Estimation of the elastic scattering rates

We estimate the elastic dipolar scattering rates of dimer–tetramer and tetramer–tetramer collisions. The scattering rate coefficient is given by $\beta = \sigma v$, where $v = \sqrt{8k_B T/\pi\mu}$ denotes the average relative velocity and $\sigma$ denotes the cross-section. In the regime of large dipole moment $E > \hbar^6/\mu^3 d_1^2 d_2^2$, the cross-section $\sigma$ can be estimated using the semiclassical formula given by[51]

$$\sigma = \frac{2}{3} \frac{d_1 d_2}{\epsilon_0 \hbar} \sqrt{\frac{\mu}{2E}}. \qquad (2)$$

Here $d_1$ and $d_2$ are the dipole moments of the two colliding particles, $\mu$ is the reduced mass and $E$ is the kinetic energy. We neglect the effect of a small ellipticity $\xi$ and estimate the effective dipole moment of the dimers to be $d_0/\sqrt{12(1 + (\Delta/\Omega)^2)}$. The dipole moment of tetramers is roughly twice as large as that of dimers. With that, the above formula provides an estimation for the elastic scattering rates to be $9.7 \times 10^{-9}$ cm$^3$ s$^{-1}$ for dimer–tetramer and $1.9 \times 10^{-8}$ cm$^3$ s$^{-1}$ for tetramer–tetramer. This implies that tens of elastic collisions can occur within the lifetime of tetramers.

### Lifetime analysis

For the measurements in time of flight, we verify in the absence of tetramers that the two-body loss between dimers is negligible during the hold time. Thus we fit an exponential decay with a constant offset given by the unpaired dimer number $N(t) = 2N_T e^{-\Gamma t} + N_D$. The offset $N_D$ is extracted from the data with ellipticity over 8°, in which the number undergoes a fast initial decay and stays constant afterwards.

To investigate the collisional stability of tetramers, we also assess their lifetimes while the dipole trap remains active. Our observations indicate a combined one-body and two-body loss of the detected dimer number, and we confirm that the two-body loss arises from dimer–dimer collisions. Apart from the data near the collisional threshold $\xi = 5(1)°$, in which in-trap measurements are influenced by thermal dissociation, we do not detect notable additional loss of tetramers in in-trap measurements compared with those in time-of-flight experiments. The deduced inelastic collision rates are consistent with zero within the error bar. We estimate that more than ten elastic collisions can occur throughout the lifetime of tetramers, which suggests that collisions with tetramers are predominantly elastic.

For measurements in a trap, we ramp up the trap depth by 50% simultaneously with the association, to compensate for the force from the inhomogeneous microwave field. The spatially varying microwave changes the dressed state energy, and thus exerts a force on the molecules that lowers the trap depth and leads to additional loss in the trapped lifetime measurements.

We first measure the total number of tetramers and dimers, and then do a comparison measurement in which we remove the tetramers as described in the main text. As shown in Extended Data Fig. 3a, we observe a two-body decay in the dimer number, in contrast to the time-of-flight measurements. To account for this background loss, we first determine the two-body loss rate $\Gamma_2$ and the initial dimer number $N_{D,0}$ from the comparison measurement and then perform a fit of one-body plus two-body decay in which we fix $\Gamma_2$ and $N_{D,0}$. The fit function is given by $N_D(t) = 2N_{T,0} e^{-\Gamma t} + N_{D,0}/(1 + \Gamma_2 t)$. Extended Data Fig. 3b,c

shows that the tetramer decay in trap and in free space are similar. The extracted decay rates differ by $9(9) \times 10^1$ Hz, which we use to obtain an upper bound for the inelastic scattering rate coefficients. By assuming that the additional loss is either purely dimer–tetramer or tetramer–tetramer, we estimate the upper bounds of their inelastic collision rate coefficients to be $2(2) \times 10^{-10}$ cm$^3$ s$^{-1}$ and $9(9) \times 10^{-10}$ cm$^3$ s$^{-1}$, respectively. Both values are consistent with zero within the error bar. Even for the worst-case estimation, the inelastic collision rate coefficients remain orders of magnitude lower than the estimated elastic dipolar scattering rate coefficients.

The lifetime of the long-range field-linked tetramers is much longer than that observed in polyatomic Feshbach molecules, which are either short lived (<1 μs) (ref. 22) or unstable in the presence of an optical trap[37]. These features make them a promising candidate for realizing a BEC of polyatomic molecules. Using the resonance at circular polarization, the improved shielding increases the tetramer lifetime to hundreds of milliseconds. As our experiments suggest that they are stable against dimer–tetramer collisions, it seems promising to evaporatively cool tetramers to lower temperatures[52].

### Association timescale analysis

We apply the following double-exponential fit to the tetramer number as a function of ramp time $t$ in Extended Data Fig. 2a

$$N_T(t) = N_0(1 - e^{-t/\tau})e^{-t_T/\tau_T}, \qquad (3)$$

where $\tau$ gives the timescale for association and $\tau_T$ gives the timescale for tetramer decay. The time $t_T \approx 0.4(t + t_{disso})$ is the time at which the ramp is above the field-linked resonance, which is about a factor of 0.4 of the association time $t$ and the dissociation time $t_{disso} = 0.5$ ms. We extract $\tau = 0.3(1)$ ms and $\tau_T = 2(1)$ ms.

### Hyperfine transitions in the modulation spectra

We observe the effects of the hyperfine structure of NaK molecules in the modulation spectra. When we modulate the ellipticity of the microwave by phase modulation, we generate two sidebands that are offset from the carrier by the modulation frequency $\nu$. When $\nu$ matches the ground- or excited-state hyperfine splitting of the dimer, a two-photon hyperfine transition occurs. In Extended Data Fig. 4b, we map out the transition spectrum by Landau–Zener sweeps, in which the modulation frequency is ramped from one data point to the next. If a sweep is performed over a hyperfine transition, molecules are transferred to another hyperfine state causing a depletion of the detected number of dimers. We observe three main hyperfine transitions from 2 kHz to 200 kHz and a few weaker ones. We verify that these transitions are not affected by changes in the ellipticity, which confirms that they are not related to the tetramer states. To obtain a clear spectrum, when measuring the dissociation spectrum, we use a small modulation amplitude to minimize power broadening and ensure that we avoid measuring near these transitions.

### Tetramer dissociation spectrum analysis

For modulation dissociation, we first create tetramers at $\xi = 8(1)°$ using electroassociation, then modulate the ellipticity for 2 ms to dissociate them. Meanwhile, we turn off the trap to suppress further association of dimers. Afterwards, we remove the remaining tetramers and let the dissociated dimers expand for another 6 ms before absorption imaging.

In addition to the hyperfine transitions mentioned above, the association of background dimers into tetramers also affects the measurement of the dissociation spectrum. However, it is worth noting that the association spectra are considerably narrower than the dissociation spectrum, and their influence can be mitigated by using a small modulation amplitude. To provide evidence for this, we present a comparative measurement in Extended Data Fig. 4a, conducted under identical

experimental conditions, except that the ellipticity ramp is as fast as 0.5 μs so that no tetramers are formed. Note that the modulation time is much shorter than for the association spectra in Fig. 2a. The observed constant background in this measurement demonstrates that the frequencies at which we measure the dissociation spectrum remain unaffected by hyperfine transitions or association.

We fit the dissociation spectrum with a dissociation line shape that is similar to the one used to describe the dissociation of Feshbach molecules[39]

$$N_T(\nu) \propto \Theta(\nu - E_b/h)\frac{\sqrt{\nu - E_b/h}}{\nu^2 + \gamma^2/4}, \qquad (4)$$

where $\Theta(\nu - E_b/h)$ is the step function and $\gamma = 20(7)$ kHz accounts for the broadening of the signal.

### Imaging method for the dissociated tetramers

Here we describe the measurement in Fig. 4b–d. We turn off the trap after the electroassociation and image the cloud after 4.5 ms of expansion time. To image the molecules, we ramp the ellipticity back to circular to rapidly dissociate the tetramers in 0.3 ms, then turn off the microwave and reverse the stimulated Raman adiabatic passage to transfer the dimers to the Feshbach molecule state. Finally, we separate the bound atoms using magnetodissociation, directly followed by absorption imaging of the atoms to minimize additional cloud expansion from residual release energy of the tetramer and Feshbach molecule dissociation.

### Angular distribution of the dissociation patterns

We average along the radial direction of the dissociation patterns to obtain their angular distribution, as shown in Extended Data Fig. 5. The distribution of the average optical density shows a sinusoidal oscillation, which matches the $p$-wave symmetry. We extract the orientation angle $\phi_0$ by fitting a function proportional to $1 + c\cos(2(\widetilde{\phi} - \phi_0))$, where $\widetilde{\phi}$ is the angle relative to the horizontal axis of the image and $c$ accounts for the finite contrast.

### Tetramer lifetime at circular polarization

The lifetime of the tetramers can be improved by shifting the field-linked resonance towards circular polarization, in which the microwave shielding is more efficient. With circular polarization, two nearly degenerate tetramer states emerge above the field-linked resonance at Rabi frequency $\Omega = 2\pi \times 83$ MHz and $\Omega = 2\pi \times 85$ MHz, which corresponds to the two $p$-wave channels with angular momentum projection $m = 1$ and $m = -1$, respectively, as shown in Extended Data Fig. 6. For the $m = 1$ state, the lifetime at binding energy $E_b < h \times 4$ kHz exceeds 100 ms. In comparison, we show the decay rate for $\xi = 5°$ for which the resonance occurs at $\Omega = 2\pi \times 28$ MHz. For the same binding energy, the lifetime is 10 times shorter than that for the $m = 1$ state because of the smaller Rabi frequency.

### Rovibrational excitations of field-linked tetramers

We investigate only the first field-linked bound state in the current experiment. At higher ellipticities and Rabi frequencies, the potential is deep enough to hold more than one bound state, which corresponds to the rovibrational excitation of the tetramers. For vibrational (rotational) excitations, the radial (axial) wavefunction of the constituent dimers has one or more nodes[53]. These excited field-linked states have more complex structures, which can be probed similarly with microwave-field modulation.

### Field-linked states of polyatomic molecules

Here we discuss the applicability of field-linked resonances to complex polyatomic molecules. For molecules in which the dipole moment is orthogonal to one of the axes of inertia, the same calculation can be

performed within the corresponding rotational subspace, as shown in ref. 10 for CaOH and SrOH. For more complex molecules in which the body-frame dipole moment is not orthogonal to any of the three axes of inertia, the microwave can induce the π transition between the ground state and the $m_J = 0$ rotational excited state. However, this detrimental π coupling can be suppressed by applying a d.c. electric field to shift the $m_J = 0$ state away from the $m_J = \pm 1$ states, so that the microwave can be off-resonant to the π transition, as shown in ref. 54. With that, a similar analysis of field-linked resonances can be applied.

## Theory

We apply coupled-channel calculations to study the scattering of molecules governed by the Hamiltonian $\hat{H} = -\nabla^2/M + \sum_{j=1,2} \hat{h}_{in}(j) + V(\mathbf{r})$, where the reduced Planck constant $\hbar = 1$.

The dynamics of a single molecule is described by the Hamiltonian $\hat{h}_{in} = B_{rot}\mathbf{J}^2 + \Omega e^{-i\omega_0 t} |\xi_+\rangle \langle 0,0|/2 + \text{h.c.}$ with the rotational constant $B_{rot} = 2\pi \times 2.822$ GHz. Here, we focus only on the lowest rotational manifolds ($J = 0$ and 1) with the four states $|J, M_J\rangle = |0, 0\rangle, |1, 0\rangle$ and $|1, \pm 1\rangle$, where $M_J$ denotes the projection of angular momentum with respect to the microwave wavevector. The microwave field of frequency $\omega_0$ and the ellipticity angle $\xi$ couples $|0, 0\rangle$ and $|\xi_+\rangle \equiv \cos\xi |1, 1\rangle + \sin\xi |1, -1\rangle$ with the Rabi frequency $\Omega$. In the interaction picture, the eigenstates of $\hat{h}_{in}$ are $|0\rangle \equiv |1, 0\rangle$, $|\xi_-\rangle \equiv \cos\xi |1, -1\rangle - \sin\xi |1, 1\rangle$, $|+\rangle \equiv u|0, 0\rangle + v|\xi_+\rangle$ and $|-\rangle \equiv u |\xi_+\rangle - v |0, 0\rangle$, and the corresponding eigenenergies are $E_0 = E_\xi = -\Delta$ and $E_\pm = (-\Delta \pm \Omega_{eff})/2$, where $u = \sqrt{(1 + \Delta/\Omega_{eff})/2}$ and $v = \sqrt{(1 - \Delta/\Omega_{eff})/2}$ with $\Delta > 0$ being the blue detuning and $\Omega_{eff} = \sqrt{\Delta^2 + \Omega^2}$ the effective Rabi frequency.

The interaction of two molecules $V(\mathbf{r}) = V_{dd}(\mathbf{r}) + V_{vdW}(\mathbf{r})$ contains the dipolar interaction

$$V_{dd}(\mathbf{r}) = \frac{d^2}{4\pi\epsilon_0 r^3}[\hat{\mathbf{d}}_1 \cdot \hat{\mathbf{d}}_2 - 3(\hat{\mathbf{d}}_1 \cdot \hat{\mathbf{r}})(\hat{\mathbf{d}}_2 \cdot \hat{\mathbf{r}})], \tag{5}$$

and the van der Waals interaction $-C_{vdW}/r^6$ ($C_{vdW} = 5 \times 10^5$ arbitrary units; ref. 55). We can project the Schrödinger equation in the two-molecule symmetric subspace $\mathcal{S}_7 \equiv \{|\alpha\rangle\}_{\alpha=1}^7 = \{|+, +\rangle, |+, 0\rangle_s, |+, \xi_-\rangle_s, |+, -\rangle_s, |-, 0\rangle_s, |-, \xi_-\rangle_s, |-, -\rangle\}$ as $\sum_{\alpha'} \hat{H}_{\alpha\alpha'}\psi_{\alpha'}(\mathbf{r}) = E\psi_\alpha(\mathbf{r})$, where $|i, j\rangle_s = (|i, j\rangle + |j, i\rangle)/\sqrt{2}$ is the symmetrization of $|i, j\rangle$. Under the rotating wave approximation, the Hamiltonian reads

$$\hat{H}_{\alpha\alpha'} = \left(-\frac{\nabla^2}{M} + \mathcal{E}_\alpha\right)\delta_{\alpha\alpha'} + V_{\alpha\alpha'}(\mathbf{r}), \tag{6}$$

where $\mathcal{E}_\alpha = \{0, -\frac{1}{2}(\Delta + \Omega_{eff}), -\frac{1}{2}(\Delta + \Omega_{eff}), -\Omega_{eff}, -\frac{1}{2}(\Delta + 3\Omega_{eff}), -\frac{1}{2}(\Delta + 3\Omega_{eff}), -2\Omega_{eff}\}$ are asymptotic energies of seven dressed states with respect to the highest dressed state channel $|1\rangle$ and $V_{\alpha\alpha'}(\mathbf{r}) = \langle\alpha| V(\mathbf{r}) |\alpha'\rangle$.

To obtain the binding energy and the decay rate of the tetramer in the dressed state $|1\rangle$, we consider a pair of molecules with incident energy $\mathcal{E}_2 < E < \mathcal{E}_1$, the angular momentum $l$ and its projection $m$ along the $z$-direction. We use the log-derivative method[56] to numerically solve the Schrödinger equation in the angular momentum basis, that is, $\psi_\alpha(\mathbf{r}) = \sum_{lm} \psi_{\alpha lm}(r) Y_{lm}(\hat{r})/r$, where the loss induced by the formation of a four-body complex is characterized using the absorption boundary condition at $r_a = 48.5a_0$. By matching the numerical solution $\psi_{\alpha lm}(r)$ with the exact wavefunction in the asymptotic region $r > R_c$, we obtain the scattering amplitudes $f_{\alpha lm}^{\alpha'l'm'}$ and the scattering cross sections $\sigma_{\alpha lm}^{\alpha'l'm'}$ from the channel ($\alpha lm$) to the channel ($\alpha'l'm'$). All results are convergent for $(l, |m|) > 7$ and $R_c > 5 \times 10^4 a_0$. We note that a different position of the absorption boundary (for example, $r_a = 32a_0$ and $r_a = 64a_0$) does not affect the result because the wavefunction has a negligible component inside the shielding core.

Without loss of generality, we concentrate on the cross-section $\sigma_{210}^{210}$ of the incident and outgoing molecules in the channel (210). When the incident energy is resonant with the tetramer state, a peak appears in the cross-section $\sigma_{210}^{210}$, where the width of the peak is the decay rate of the tetramer. The cross-section $\sigma_{210}^{210}$ quantitatively agrees with the lineshape

$$\sigma(E) = \frac{2\pi}{k_2^2} |ig^2 G(E) + S_{bg} - 1|^2, \tag{7}$$

where $G(E) = 1/(E - E_b + i\Gamma/2)$ is the tetramer propagator, $k_2 = \sqrt{M(E - \mathcal{E}_2)}$ and $S_{bg}$ are the incident momentum and the background scattering amplitude of molecules in the dressed state channel $|2\rangle$, respectively. By fitting $\sigma_{210}^{210}$ and $\sigma(E)$, we obtain the binding energy $E_b$ and the decay rate $\Gamma$ of the tetramer. We remark that for the incident and outgoing molecules in other channels $\alpha \approx 3-7$, the propagator $G(E)$ in equation (7) does not change. Therefore, fitting $\sigma_{\alpha lm}^{\alpha'l'm'}$ in a different scattering channel leads to the same binding energy $E_b$ and decay rate $\Gamma$.

For a tetramer with a small decay rate, its wavefunction $\psi_b(\mathbf{r})$ can be obtained by solving the Schrödinger equation $H_{eff}\psi_b(\mathbf{r}) = \bar{E}_b\psi_b(\mathbf{r})$. The single-channel model $H_{eff} = -\Delta^2/M + V_{eff}(\mathbf{r})$ is determined by the effective potential[15]

$$V_{eff}(\mathbf{r}) = \frac{C_6}{r^6}\sin^2\theta\{1 - \mathcal{F}_\xi^2(\phi) + [1 - \mathcal{F}_\xi(\phi)]^2\cos^2\theta\}$$
$$+ \frac{C_3}{r^3}[3\cos^2\theta - 1 + 3\mathcal{F}_\xi(\phi)\sin^2\theta] \tag{8}$$

for two molecules in the dressed state channel $|1\rangle$, where $\mathcal{F}_\xi(\phi) = -\sin 2\xi \cos 2\phi$, $\theta$ and $\phi$ are the polar and azimuthal angles of $\mathbf{r}$. The strength $C_3 = d^2/[48\pi\epsilon_0(1 + \delta_r^2)]$ of the dipole-dipole interaction depends only on the relative detuning $\delta_r = |\Delta|/\Omega$, whereas the $C_6$ term describes an anisotropic shielding potential that prevents destructive short-range collisions. Using the B-spline algorithm, we obtain the binding energy $\bar{E}_b$ and the wavefunction $\psi_b(\mathbf{r}) \approx Y_{1-}(r)\varphi_1(r)/r$ of the first tetramer bound state, where $Y_{1-}(r) = (Y_{11}(r) - Y_{1-1}(r))/\sqrt{2}$. The binding energies $\bar{E}_b$ and $E_b$ obtained from the single-channel model and the seven-channel scattering calculation agree with each other quantitatively for small $\xi$ and $\Omega$. For the largest $\xi$ and $\Omega$ in Fig. 1, the relative error of $\bar{E}_b$ is less than 30%. The tetramer wavefunction in the momentum space is the Fourier transform $\psi_b(\mathbf{k}) = \int d\mathbf{r} e^{-i\mathbf{k}\cdot\mathbf{r}}\psi_b(\mathbf{r})/(2\pi)^{3/2}$ of $\psi_b(\mathbf{r})$.

For the modulation dissociation, the transition probability $p_{\mathbf{k}}$ to the momentum state $\mathbf{k}$ is determined by the coupling strength $g_{\mathbf{k}} = \int d\mathbf{r}\psi_{\mathbf{k}}^*(\mathbf{r})\partial_\xi V_{eff}(\mathbf{r})\psi_b(\mathbf{r})$. Here, $\psi_{\mathbf{k}}(\mathbf{r})$ represents the wavefunction of the scattering state. The coupling strength $g_{\mathbf{k}}$ is primarily influenced by $Y_{1-}(\hat{k})$, which characterizes the angular distribution of $\psi_b(\mathbf{k})$. This dominance arises because $\partial_\xi V_{eff}$ maintains mirror symmetry with respect to the $x$–$y$ plane. Therefore, by measuring $p_{\mathbf{k}} \approx |Y_{1-}(\hat{k})|^2$, we can effectively probe the angular dependence of the tetramer state in the momentum space.

## Data availability

The experimental data that support the findings of this study are available from the corresponding authors upon request.

## Code availability

All relevant codes are available from the corresponding authors upon request.

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

**Acknowledgements** We thank G. Quéméner for stimulating discussions. We acknowledge support from the Max Planck Society and the Deutsche Forschungsgemeinschaft under Germany's Excellence Strategy—EXC-2111—390814868 and under grant no. FOR 2247. F.D., T.S. and S.Y. acknowledge support from the National Key Research and Development Program of China (grant no. 2021YFA0718304), National Natural Science Foundation of China (grant nos. 11974363 and 12274331) and CAS Project for Young Scientists in Basic Research (grant no. YSBR-057).

**Author contributions** All authors contributed substantially to the work presented in this paper. X.-Y.C. and S.B. carried out the experiments and together with S.E. and A.S. improved the experimental setup. X.-Y.C., S.E. and S.B. analysed the data. F.D., T.S. and S.Y. performed the theoretical calculations. T.H., I.B. and X.-Y.L. supervised the study. All authors worked on the interpretation of the data and contributed to the final paper.

**Funding** Open access funding provided by Max Planck Society.

**Competing interests** The authors declare no competing interests.

**Additional information**
**Correspondence and requests for materials** should be addressed to Tao Shi or Xin-Yu Luo.

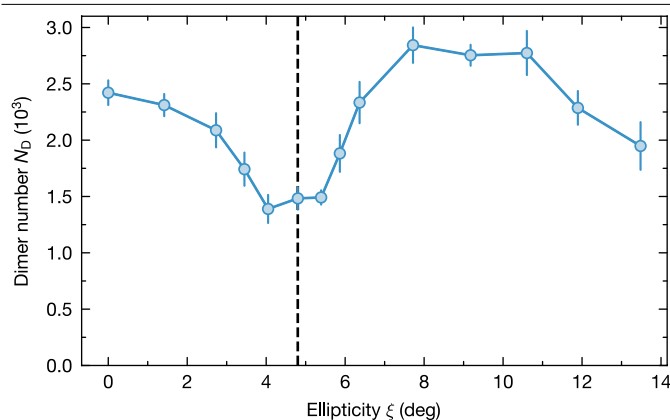

**Extended Data Fig. 1 | Dimer loss near the FL resonance.** Number $N_D$ of remaining dimers as a function of ellipticity $\xi$. The hold time is 100 ms. The Rabi frequency of the microwave field is $\Omega = 2\pi \times 29(1)$ MHz and the detuning is $\Delta = 2\pi \times 9.5$ MHz. The error bars represent the standard error of the mean of four repetitions.

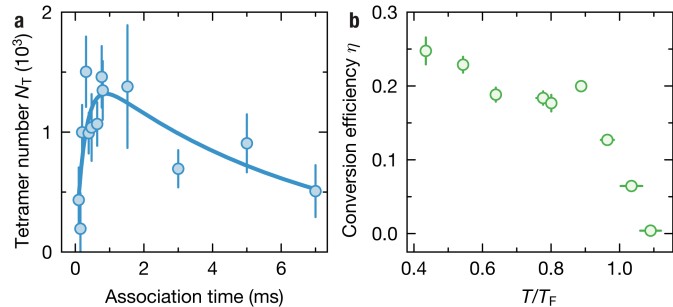

**Extended Data Fig. 2 | Conditions for efficient electroassociation. a**, Tetramer number $N_T$ as a function of the association time. The solid blue line is a fit to a double exponential function, which captures the formation and decay of the tetramers (Methods). The error bars represent the standard error of the mean of eight repetitions. **b**, Conversion efficiency $\eta$ as a function of the initial $T/T_F$ of the dimer gas. We use a ramp speed of $7°\,ms^{-1}$ for the electroassociation. The initial $T/T_F$ are extracted separately, without performing electroassociation. The error bars represent the standard error of the mean of four repetitions.

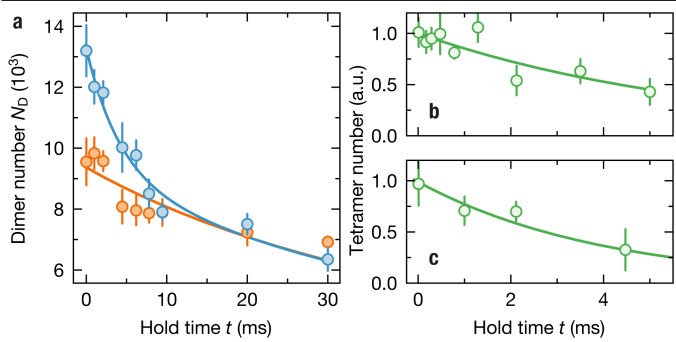

**Extended Data Fig. 3 | Tetramer lifetime in trap and in time-of-flight.**
**a**, Example loss of the molecule number with (orange) and without (blue)
removal of tetramers in the trap at $\xi = 7(1)°$. **b**, Normalized tetramer decay
measured in time-of-flight at the same ellipticity $\xi$. **c**, Extracted tetramer number
from the data in **a**. No notable additional loss is observed compared to **b**. The
error bars represent standard error of the mean of ten repetitions.

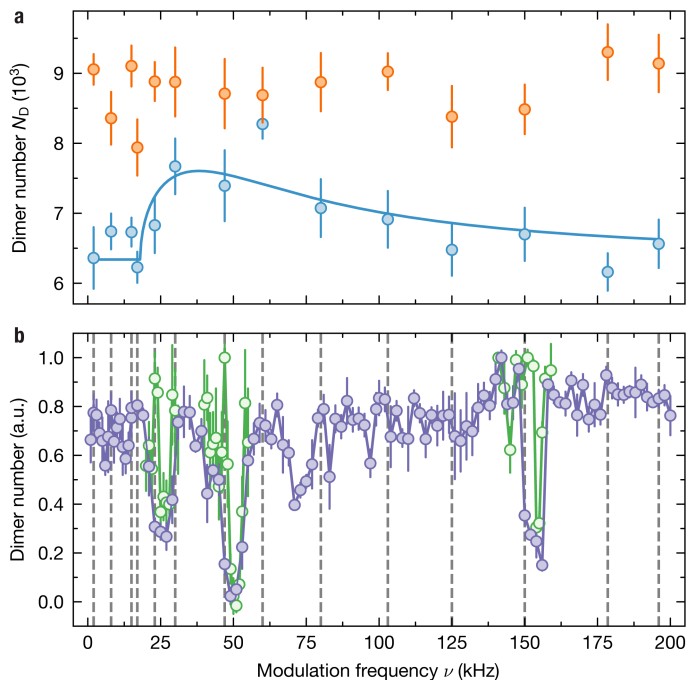

**Extended Data Fig. 4 | Hyperfine transitions of NaK molecules in the modulation spectra. a**, Tetramer dissociation spectrum (blue) compared to background loss (orange). The background loss is measured under the same condition as the dissociation spectrum, except with a fast ramp over the FL resonance so that no tetramers are formed. The absence of loss in the background measurement suggests that the dissociation spectrum is not affected by either the hyperfine transitions or the association. This is ensured by using a small modulation amplitude of 1. 4° over a duration of 2 ms and by taking the measurements away from known hyperfine transitions. The error bars represent standard error of the mean of ten repetitions. **b**, Hyperfine spectrum measured with Landau–Zener sweeps in the modulation frequency and a modulation amplitude of 11° (purple) and 3. 6° (green). The transitions with the larger modulation amplitude is power broadened compared to the lower amplitude ones. The modulation frequency that we use for the tetramer dissociation spectrum are marked as vertical dashed lines. The error bars represent standard error of the mean of four repetitions.

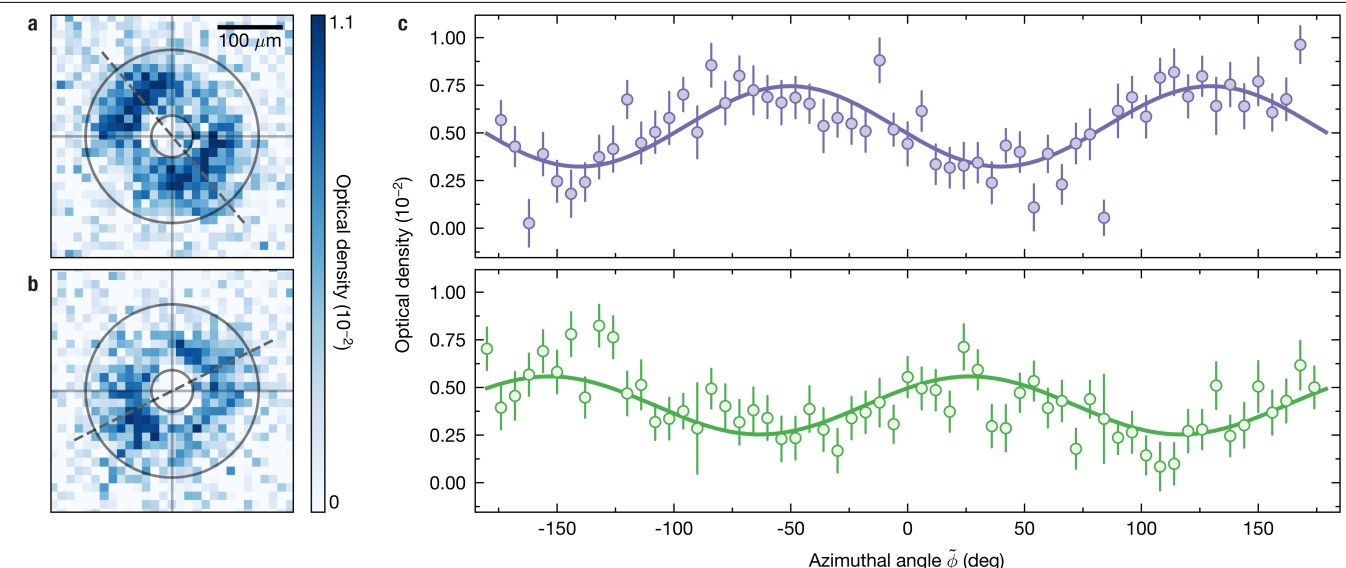

**Extended Data Fig. 5 | Tetramer dissociation patterns and their angular distribution. a,b**, Images of the modulation-dissociated tetramers at different microwave field orientations. For each image, we average over the areas between the two circles to obtain the angular distribution of the optical density as shown in **c**. The dashed line marks the extracted orientation of the elliptical microwave polarization, which is $\phi_0 = -50(4)°$ (**a**) and $\phi_0 = 27(4)°$ (**b**). **c**, Angular distribution of the optical density. The upper and lower panels show the data corresponding to the images of **a** and **b**, respectively.

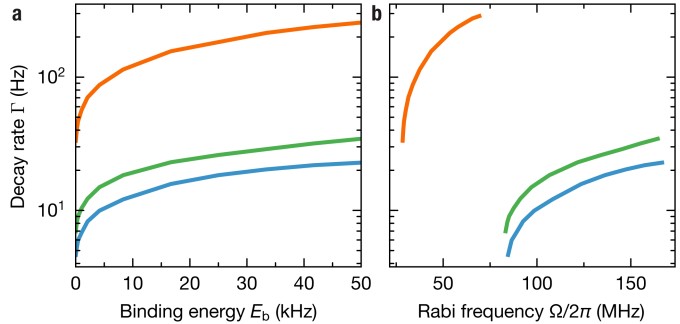

**Extended Data Fig. 6 | Theoretical tetramer decay rate. a,b**, Decay rate Γ as a function of the binding energy (**a**) and Rabi frequency (**b**). Each curve in the figures represents a calculation with a fixed $\xi$ while varying the Rabi frequencies, resulting in a range of binding energies. The blue (green) curve corresponds to circular polarization with angular momentum projection $m = 1$ ($m = -1$). The orange curve corresponds to $\xi = 5°$.