## [Peer Review File · Nature]

Manuscript Title: Ultracold Field-Linked Tetratomic Molecules

Reviewer Comments & Author Rebuttals

Reviewer Reports on the Initial Version:

Referees' comments:

Referee #1 (Remarks to the Author):

The manuscript describes experiments where weakly bound tetramer (dimer-dimer) molecules are formed in the sub microKelvin regime. The dimer-dimer binding energy is measured and in a tour de force set of measurements the complex system is studied and optimized. I feel that this is a very strong paper and is at the same level importance to science as the first weakly bound dimer (atom-atom) work done more than a decade ago. This manuscript is very appropriate for publication in Nature. (I will also note that the mastering of this highly complex system using the tools of atomic physics is of itself noteworthy, at least to those from the chemist community.)

The manuscript is excellently written with only a few opportunities for improving communication to the specialized and broader audiences. I will outline them below and explain my reasoning why they deserve attention.

The achievement here is the creation of weakly bound dimer-dimer molecules. This is a great achievement, worthy of publication in Nature. However, many of the applications for polyatomic molecules (as outlined in the references given in the manuscript) will rely on much more deeply bound molecules, typically in the electronic ground state. The authors outline a future path towards this, in the conclusion. Without taking anything away from the paper - and at the same time elucidating the field for the reader - I suggest that "weakly bound" be put in right before "tetratomic (NaK)₂" in the abstract. This will put up front this fact and avoid a possible backlash, especially from chemists who will be a large part of the audience who will click on the link to this paper.

Another opportunity for educating the audience is in the second paragraph where laser cooling of polyatomic molecules is mentioned. At this point in time, this is a very open question (as is clear from the papers from UCLA, Isaev and Berger, and the Doyle group). In particular, it there are predictions that even large molecules may need only a limited number of lasers. This is a frontier chemistry/physics question. I suggest replacing that sentence with something like "Laser cooling of larger polyatomic molecules is an area of active research[Refs]. Although symmetric top molecules have been laser cooled in one dimension [ref], it remains to be seen how efficient laser cooling of large (tetramer and larger) molecules will be in three dimensions and into the submicroKelvin regime." The point is not particularly to toot the horn of that field, but, rather, clearly and succinctly delineate the status of the field and the prospects.

Page 4, "in average"  "on average".

Referee #2 (Remarks to the Author):

This paper reports on extraordinary results regarding the possibility to assemble two ultracold polar molecules into a bound state which is induced by an ambient microwave field, thus forming a weakly-bound complex of a pair of molecules. This opportunity was prepared by previous work of the same group, related to microwave shielding of intermolecular interactions, which thus prevent two ultracold molecules to decay into some unknown products due to short-range interactions. This possibility of observing such a bound state, hereafter referred to as "electroassociation" (a name proposed in [7], which in passing, may not be the most appropriate; also this reference must be updated as the paper just appeared in PRL), relies on the observation of corresponding resonances by the same group (ref. 11, in Nature too).

The reported experimental results are all extremely impressive, and precise, and the variety of tests proposed by the authors to cross-check their observations is exquisite.

Therefore, I certainly recommend the publication of this paper. However, I have a couple of important remarks, that should be taken in account by the authors, prior to final approval.

1) While the title refers to "tetraatomic molecules", which reflects the present study in a general sense, I find misleading to compare the present weakly-bound complex of two polar molecules to a generic "polyatomic molecule", as suggested in the abstract, and in the introduction when invoking references related to the attempts cool down "genuine" polyatomic molecules (see refs 29-34). The present system is really different such strongly-bound polyatomic molecules, and this should be properly stated. Correspondingly the sentence "Here we demonstrate a novel and general approach to form ultracold polyatomic molecules by electroassociation of smaller polar molecules" is also misleading, the proposed method is general to create weakly bound complexes with the help of electromagnetic fields, but not to create polyatomic molecules in the sense of refs 29-34. There are really two different physical objects, and this must be properly assessed. As a further argument, we see in Fig.1d that the binding energy of these tetramers is orders of magnitude smaller than the binding energies of the polyatomic molecules of refs 29-34.

2) At the bottom of p1 and top of p2, the sentence "We create ultracold tetraatomic (NaK)₂ molecules from pairs of microwave dressed fermionic NaK molecules" suffers, in my opinion, from the will to compress the text as much as possible in this introductory part. Indeed, the molecules by themselves are not dressed by the mw field. These are the mw-dressed states resulting from the mw-dressed Hamiltonian of the two molecules.

3) The next sentence, "This approach benefits from the universality of FL resonances and can be applied to any molecule with a sufficiently large dipole moment" also seems to me a bit overstated. This statement also appears in the conclusion as "Thanks to the universality of field linked resonance, our approach can be generalized to a wide range of polar molecules, including more complex polyatomic molecules". Indeed, the dynamics of such field-linked states results from the interplay of dipole-dipole interaction, of the internal rotation of the molecules, and of their mutual rotation (or say, partial wave). I am not sure if a molecule more complex than a diatomic one, with several axis of inertia, would lead to the same results. I admit that I am a bit reluctant with the somewhat too easy use of the word "universality" in this research field.

4) Figure 1 requires little bit more care: the horizontal axis of Fig.1b, with so few ticks, suggests to the quick eye that the created tetraatomic molecule is characterized by some sort of short-range potential well. I recommend to the authors to display ticks every 1000a₀, so that the image is immediately clear to the reader. Also in the caption the sentence "Interaction potentials

between two dimers approaching along the long axis of the microwave field at $\xi = 0^\circ$ (blue) and $\xi = 14^\circ$ (orange)" is confusing. First, the ellipticity (well defined in the text) could be here understood as the angle between the collision axis and the electric field axis. Second, if I understood well such an image assumes a given direction in the lab space of the collision axis, which is not the case in real life. The full picture has to be built by averaging over all the possible angles of the collision axis. It is not clear to me if this is the approach which has indeed been considered here, which results are displayed in Fig.2 for instance "without free parameters".

5) Finally I have some concerns in understanding the theory part in "METHODS". First - a minor point- the notation J (capital j) is used for the angular momentum of the individual molecules, where the convention would be to use j (small j) instead, thus keeping J (capital j) for the total angular momentum of the complex. Next, after eq. 5, I do not see the value of the van der Waals coefficient which is used in the model. What is the meaning of the so-called "effective Rabi frequency", beyond its definition? Is it describing indeed the off-diagonal terms of the mw-dressed hamiltonian of the two interacting molecules? I understand that the calculations are performed in the uncoupled basis in the lab frame, as particular values of ℓ (the "angular momentum", which I suspect is the mechanical rotation of the two dimers) and its projection m . But what about the total angular momentum of the complex, and its total projection? What is the range of values of ℓ and m which are considered in the calculations, to check the convergence of the calculations? How do the results depend on the absorption boundary condition at $r=48.5a_0$? Does this value result from a specific adjustment? Should we understand eq.7 as the elastic cross section in a given channel, which exhibits a peak due to the tetramer resonance? To conclude, I still have the impression -which may be clarified by the authors- that the theoretical parameters relies on some parameters, so that the statement above that it is used "without free parameters" may be overstated.

Author Rebuttals to Initial Comments:

Response to the referee reports

First of all, we thank the referees for reviewing our work and their valuable comments and positive recommendations. We are delighted to read that the referees found our results a “tour de force” and “extremely impressive”. We have revised the manuscript according to the suggestions of the referees. We have also shortened the main text to meet the length limit allowed by Nature. In the following response, we address the concerns raised by the referees point-by-point and explain the modifications we have undertaken as a result.

We also submit a main-diff pdf file showing all changes and summarize the major modifications in the list below.

1. We shortened the abstract by removing the sentence “The measured binding energy and lifetime agree well with parameter-free calculations, which outlines pathways to further increase the lifetime of the tetramers.”
2. We added “weakly bound” in key statements in the abstract, introduction, and conclusions.
3. In the discussion section, we explicitly mentioned CaOH and SrOH as examples of polyatomic molecules where FL resonances have been predicted. We also added a section “FL states of polyatomic molecules” to the Methods section.
4. We adopt Referee 1’s suggestion on laser cooling in the introduction. We replaced “Direct laser cooling ... limiting efficient photon scattering.” to “Laser cooling of larger polyatomic molecules is an area of active research. Although symmetric top molecules have been laser cooled in one dimension, it remains to be seen how efficient laser cooling of large (tetraatomic or larger) molecules will be in three dimensions and whether temperatures below the submicrokelvin regime can be achieved.”
5. We moved the section “Conditions for efficient electroassociation” and the corresponding figure to Methods.
6. We moved the discussion about tetramer collisions to the section “Lifetime analysis” in Methods.
7. We moved the imaging method of the dissociated tetramers to the section “Imaging method for the dissociated tetramers” in Methods.
8. We moved the discussion about higher-order FL states to the section “Rovibrational excitations of FL tetramers” in Methods.
9. We added a citation to [Phys. Rev. Research 4, 013235 (2022)] in the introduction, and two citations to [Phys. Rev. A 87, 022706 (2013)] and [Phys. Rev. A 101, 042702 (2020)] in Methods. The total number of references in the main text remains below 50.
10. We updated the citation to [Phys. Rev. Lett. 131, 043402 (2023)], as it has been published recently.
11. We corrected the following typos: “substracting” → “subtracting”, “in average” → “on average”.
12. We corrected a typo in “For the PES of (NaK)₂ molecules, there are seven energy minima...”, which should instead be “three”.
13. We updated subfigure **f** of the last figure in the main text with the theoretical dissociation pattern.
14. We added the model number of the 100 W amplifiers in Methods.

RESPONSE TO REPORT FROM REFEREE #1

The manuscript describes experiments where weakly bound tetramer (dimer-dimer) molecules are formed in the sub microkelvin regime. The dimer-dimer binding energy is measured and in a tour de force set of measurements the complex system is studied and optimized. I feel that this is a very strong paper and is at the same level importance to science as the first weakly bound dimer (atom-atom) work done more than a decade ago. This manuscript is very appropriate for publication in Nature. (I will also note that the

mastering of this highly complex system using the tools of atomic physics is of itself noteworthy, at least to those from the chemist community.)

The manuscript is excellently written with only a few opportunities for improving communication to the specialized and broader audiences. I will outline them below and explain my reasoning why they deserve attention.

(A0) We thank the referee for the valuable comments and recommendation of publication.

The achievement here is the creation of weakly bound dimer-dimer molecules. This is a great achievement, worthy of publication in Nature. However, many of the applications for polyatomic molecules (as outlined in the references given in the manuscript) will rely on much more deeply bound molecules, typically in the electronic ground state. The authors outline a future path towards this, in the conclusion. Without taking anything away from the paper - and at the same time elucidating the field for the reader - I suggest that "weakly bound" be put in right before "tetraatomic (NaK)₂" in the abstract. This will put up front this fact and avoid a possible backlash, especially from chemists who will be a large part of the audience who will click on the link to this paper.

(A1) We agree with the referee. Indeed, field-linked molecules are "exotic" long-range and weakly bound molecules, which are distinct from more deeply-bound molecules. Therefore, we added "weakly bound" to key statements in the abstract, introduction, and conclusion.

Another opportunity for educating the audience is in the second paragraph where laser cooling of polyatomic molecules is mentioned. At this point in time, this is a very open question (as is clear from the papers from UCLA, Isaev and Berger, and the Doyle group). In particular, there are predictions that even large molecules may need only a limited number of lasers. This is a frontier chemistry/physics question. I suggest replacing that sentence with something like "Laser cooling of larger polyatomic molecules is an area of active research[Refs]. Although symmetric top molecules have been laser cooled in one dimension [ref], it remains to be seen how efficient laser cooling of large (tetramer and larger) molecules will be in three dimensions and into the submicroKelvin regime." The point is not particularly to toot the horn of that field, but, rather, clearly and succinctly delineate the status of the field and the prospects.

(A2) We thank the referee for providing a clear delineation for the status of the field of direct laser cooling of polyatomic molecules. We have adopted the referee's suggestion and changed the text accordingly.

RESPONSE TO REPORT FROM REFEREE #2

This paper reports on extraordinary results regarding the possibility to assemble two ultracold polar molecules into a bound state which is induced by an ambient microwave field, thus forming a weakly bound complex of a pair of molecules. This opportunity was prepared by previous work of the same group, related to microwave shielding of intermolecular interactions, which thus prevent two ultracold molecules to decay into some unknown products due to short-range interactions. This possibility of observing such a bound state, hereafter referred to as "electroassociation" (a name proposed in [7], which in passing, may not be the most appropriate; also this reference must be updated as the paper just appeared in PRL), relies on the observation of corresponding resonances by the same group (ref. 11, in Nature too). The reported experimental results are all extremely impressive, and precise, and the variety of tests proposed by the authors to cross-check their observations is exquisite. Therefore, I certainly recommend the publication of this paper. However, I have a couple of important remarks, that should be taken in account by the authors, prior to final approval.

(B0) We thank the referee for the positive feedback and recommendation of publication.

1) While the title refers to "tetraatomic molecules", which reflects the present study in a general sense, I find misleading to compare the present weakly bound complex of two polar molecules to a generic "polyatomic molecule", as suggested in the abstract, and in the introduction when invoking references related to the attempts to cool down "genuine" polyatomic molecules (see refs 29-34). The present system is really different such as strongly-bound polyatomic molecules, and this should be properly stated. Correspondingly the sentence "Here we demonstrate a novel and general approach to form ultracold polyatomic molecules by electroassociation of smaller polar molecules" is also misleading, the proposed method is general to create weakly bound complexes with the help of electromagnetic fields, but not to create polyatomic molecules in the sense of refs 29-34. There are really two different physical objects, and this must be properly assessed. As a further argument, we see in Fig.1d that the binding energy of these tetramers is orders of magnitude smaller than the binding energies of the polyatomic molecules of refs 29-34.

(B1) We agree with the referee that the field-linked molecules are unconventional polyatomic molecules compared to deeply bound ones. The long-range and weakly bound nature of the field-linked molecules makes them highly tunable by external fields. To distinguish them from deeply bound molecules, we add "weakly bound" to the key statements in abstract, introduction, and conclusions. As suggested by Referee 1, the term has been widely used to distinguish Feshbach molecules from ground-state molecules, where initially a similar discussion took place.

Note that the word "complex" suggested by the referee has been used for the tightly bound intermediate states in molecular collisions [J. Phys. Chem. A, 127, 3, 729-741 (2023)]. These complexes are naturally formed and subsequently decay during collisions, leading to a two-body loss of the molecular gas. Thus we do not refer to the FL molecules as complexes.

2) At the bottom of p1 and top of p2, the sentence "We create ultracold tetraatomic (NaK)₂ molecules from pairs of microwave dressed fermionic NaK molecules" suffers, in my opinion, from the will to compress the text as much as possible in this introductory part. Indeed, the molecules by themselves are not dressed by the mw field. These are the mw-dressed states resulting from the mw-dressed Hamiltonian of the two molecules.

(B2) We rewrite the sentence as "We create weakly bound ultracold tetraatomic (NaK)₂ molecules from pairs of fermionic NaK molecules in the microwave-dressed states".

3) The next sentence, "This approach benefits from the universality of FL resonances and can be applied to any molecule with a sufficiently large dipole moment" also seems to me a bit overstated. This statement also appears in the conclusion as "Thanks to the universality of field linked resonance, our approach can be generalized to a wide range of polar molecules, including more complex polyatomic

molecules". Indeed, the dynamics of such field-linked states results from the interplay of dipole-dipole interaction, of the internal rotation of the molecules, and of their mutual rotation (or say, partial wave). I am not sure if a molecule more complex than a diatomic one, with several axis of inertia, would lead to the same results. I admit that I am a bit reluctant with the somewhat too easy use of the word "universality" in this research field.

(B3) As pointed out by the referee, "the field-linked states result from the interplay of dipole-dipole interaction, of the internal rotation of the molecules, and of their mutual rotation". For polyatomic molecules whose dipole moment is orthogonal to one of the axis of inertia, the same calculation can be performed within the corresponding rotational subspace, as shown in [Phys. Rev. Lett. 121, 163402 (2018)] for CaOH and SrOH. For more complex molecules, where the body-frame dipole moment is not orthogonal to any of the three axes of inertia, the microwave can induce a π transition between the ground state and the $m_J = 0$ rotational excited state. However, this π transition can be suppressed by applying a d.c. electric field to shift the $m_J = 0$ state away from the $m_J = \pm 1$ states, so that the microwave can be off-resonant to the π transition, as shown in [Phys. Rev. A 101, 042702 (2020)]. With that, a similar analysis of FL resonance applies. We admit that, given the complexity of molecules, the properties of the field-linked molecules vary, but the same physics holds. We added a corresponding section titled "FL states of polyatomic molecules" in Methods to now include this discussion in the manuscript.

4) Figure 1 requires little bit more care: the horizontal axis of Fig.1b, with so few ticks, suggests to the quick eye that the created tetraatomic molecule is characterized by some sort of short-range potential well. I recommend to the authors to display ticks every $1000a_0$, so that the image is immediately clear to the reader. Also in the caption the sentence "Interaction potentials between two dimers approaching along the long axis of the microwave field at $= 0^\circ$ (blue) and $= 14^\circ$ (orange)" is confusing. First, the ellipticity (well defined in the text) could be here understood as the angle between the collision axis and the electric field axis. Second, if I understood well such an image assumes a given direction in the lab space of the collision axis, which is not the case in real life. The full picture has to be built by averaging over all the possible angles of the collision axis. It is not clear to me if this is the approach which has indeed been considered here, which results are displayed in Fig.2 for instance "without free parameters".

(B4) We added more ticks as suggested by the referee. We thank the referee for pointing out the confusion on the collision axis and the elliptic angle ξ . In our experiment, the ellipse of the microwave field polarization is always pointing towards a fixed direction in the lab frame, which we defined as x-axis. The wave function of the field-linked state is oriented along that direction, as illustrated in the Fig. 1a.

Figure. 1b,c show the potential curves along the x-axis as this is the most attractive direction. For theory calculations in Fig. 2, all the collisional directions are considered. We discuss the theory in more detail in (B5.2).

5) Finally, I have some concerns in understanding the theory part in "METHODS". First - a minor point- the notation J (capital j) is used for the angular momentum of the individual molecules, where the convention would be to use j (small j) instead, thus keeping J (capital j) for the total angular momentum of the complex. Next, after eq. 5, I do not see the value of the van der Waals coefficient which is used in the model. What is the meaning of the so-called "effective Rabi frequency", beyond its definition? Is it describing indeed the off-diagonal terms of the mw-dressed hamiltonian of the two interacting molecules? I understand that the calculations are performed in the uncoupled basis in the lab frame, as particular values of ell (the "angular momentum", which I suspect is the mechanical rotation of the two dimers) and its projection m . But what about the total angular momentum of the complex, and its total projection?

(B5.1) The total angular momentum of the FL tetramer is not a good quantum number, as each dressed diatomic molecule is constantly exchanging angular momentum with the microwave photons. A better quantum number is the orbital angular momentum of the relative motion of the two dimers l , which naturally arises in the partial wave expansion. The dipole-dipole interaction mixes different l states, however, the dominant scattering channel is the lowest partial wave $l = 1$ for identical fermions, i.e. p -wave scattering. The p -wave symmetry of the tetramer wave function is probed by imaging the modulation dissociated tetramers, as shown in Fig. 4c,d and Extended Data Fig. 5.

The effective Rabi frequency Ω_{eff} is the frequency of the Rabi oscillation in the presence of a detuning Δ .

What is the range of values of l and m which are considered in the calculations, to check the convergence of the calculations?

(B5.2) The calculations are performed using the coupled multi-channel scattering approach [Phys. Rev. Lett. 130, 183001 (2023)]. The idea is to project the Schrödinger equation into the coupled angular momentum basis. If the elliptic angle is zero, the total projection angular momentum m is conserved, and we focus on the lowest channel $m = \pm 1$, where the first tetramer state appears. For a finite elliptic angle, multiple m states need to be considered. For all calculations, we checked the convergence of the result by increasing the cut-off l_c and m_c of l, m . We found that all the results converge when l_c and m_c is larger than 7. We added this information to the Methods section.

How do the results depend on the absorption boundary condition at $r=48.5a_0$? Does this value result from a specific adjustment?

(B5.3) The absorption boundary condition comes from the observation that when two molecules approach short distance, there is a near unitary probability of loss, also known as *universal loss* proposed in [Phys. Rev. Lett. 104, 113202 (2010)]. The result is not sensitive to the size of the absorption boundary. One can obtain the same result by increasing or decreasing, e.g., $r = 32a_0$ or $64a_0$. Since the two-molecule wave function has a tiny distribution inside the shielding core around several hundred Bohr radius, the result does not change if the position of the absorption boundary is much smaller than the shielding core. In fact, in our previous publication [Nature 614, 59–63 (2023)], the absorption boundary was set to $r = 20a_0$.

Should we understand eq.7 as the elastic cross section in a given channel, which exhibits a peak due to the tetramer resonance?

(B5.4) The referee is correct about Eq. 7. We consider the incident molecules in the second highest dressed state channel. The first term determined by the Green's function $G(E)$ of the tetramer exhibits a peak when the incident energy is resonant with the tetramer state, while the second term $S_{\text{bg}} - 1$ describes the background scattering in the second dressed state channel. The position and width of the resonant peak determine the binding energy and the lifetime of the tetramer.

To conclude, I still have the impression -which may be clarified by the authors- that the theoretical parameters relies on some parameters, so that the statement above that it is used "without free parameters" may be overstated.

(B5.5) As discussed above, we do not include any free parameters in the theory calculation. The only parameters that enter into the theory are the dipole moment d_0 , the rotational constant B , the mass of the molecule m , and the van der Waals potential (obtained from ab-initio calculations in [Phys. Rev. A 87, 022706 (2013)]). We added a citation to the van der Waals coefficient to the Methods section.

Reviewer Reports on the First Revision:

Referees' comments:

Referee #2 (Remarks to the Author):

I thank the authors for having carefully taken in account all my concerns, and for having nicely combined the suggestions from the two referees. In particular:

- I appreciated the discussion on polyatomic molecules as a specific section, it is very enlightening.
- Having the convergence threshold of the results for R_c and ℓ is useful
- I admit that my comment regarding the "weakly bound" character of the present tetramers held also when weakly bound diatomic molecules have been created. And personally, I always named them as weakly bound :). But it is indeed a nice prospect as a first step toward more deeply-bound molecules, even if one can anticipate that the path could be far more complex.

Therefore I consider that the manuscript in its present form is suitable for publication in Nature

Author Rebuttals to First Revision:

Response to the referee reports

We thank the referees for reviewing our work and their valuable comments and positive recommendations. Here we summarize the modifications in the list below.

1. We added one citation to [Phys. Rev. A 85, 055601 (2012)]. We now cite 49 references in the main text.
2. We added one grant CAS Project for Young Scientists in Basic Research (Grant No. YSBR-057) in the acknowledgement.

RESPONSE TO REPORT FROM REFEREE #2

I thank the authors for having carefully taken in account all my concerns, and for having nicely combined the suggestions from the two referees. In particular:

- ◊ I appreciated the discussion on polyatomic molecules as a specific section, it is very enlightening.
- ◊ Having the convergence threshold of the results for R_c and ℓ is useful.
- ◊ I admit that my comment regarding the "weakly bound" character of the present tetramers held also when weakly bound diatomic molecules have been created. And personally, I always named them as weakly bound :). But it is indeed a nice prospect as a first step toward more deeply-bound molecules, even if one can anticipate that the path could be far more complex.

Therefore I consider that the manuscript in its present form is suitable for publication in Nature.

We thank the referee for their appreciative comments and recommendation for publication in Nature.